# Tissue-Resident Innate and Innate-Like Lymphocyte Responses to Viral Infection

**DOI:** 10.3390/v11030272

**Published:** 2019-03-19

**Authors:** Andrew D. Hildreth, Timothy E. O’Sullivan

**Affiliations:** Department of Microbiology, Immunology, and Molecular Genetics, David Geffen School of Medicine at UCLA, Los Angeles, CA 900953, USA; ahildret@g.ucla.edu

**Keywords:** tissue resident, innate lymphoid cell, unconventional T cell, viral infection

## Abstract

Infection is restrained by the concerted activation of tissue-resident and circulating immune cells. Recent discoveries have demonstrated that tissue-resident lymphocyte subsets, comprised of innate lymphoid cells (ILCs) and unconventional T cells, have vital roles in the initiation of primary antiviral responses. Via direct and indirect mechanisms, ILCs and unconventional T cell subsets play a critical role in the ability of the immune system to mount an effective antiviral response through potent early cytokine production. In this review, we will summarize the current knowledge of tissue-resident lymphocytes during initial viral infection and evaluate their redundant or nonredundant contributions to host protection or virus-induced pathology.

## 1. Introduction

In recent years, it is becoming evident that a complete immune response requires an evolutionary division of labor between tissue-resident and circulating responses. Innate immune cells, such as macrophages and dendritic cells (DCs), generally reside in peripheral tissues and can recognize pathogens through the expression of toll-like receptors (TLRs) to produce a wide-variety of effector molecules that can directly or indirectly limit early pathogen replication at the site of infection [1]. However, the recently described diversity and relative abundance of tissue-resident lymphocytes in peripheral organs suggests that early myeloid responses alone cannot efficiently restrain pathogen replication in infected tissues before initiation of antigen-specific T cell responses [2]. Tissue-resident lymphocytes, such as innate lymphoid cells (ILCs) and “unconventional” T cells (invariant natural killer T cells (iNKT), γδ T cells, and mucosal associated invariant T cells (MAIT)), have been shown to have critical roles during the pathogen challenge [2,3,4]. In this review, we will summarize the current knowledge of the direct and indirect roles of tissue-resident innate and innate-like lymphocytes in suppressing or promoting primary viral-mediated pathology in tissues.

## 2. Discussion

### 2.1. Innate Lymphoid Cells

ILCs consist of a heterogeneous family of lymphocytes that do not express rearranged antigen receptors, but instead express a wide variety of germline-encoded activating and inhibitory receptors [3]. ILCs can be identified in lymphoid and non-lymphoid tissues, and are enriched at epithelial barrier surfaces such as the intestine, lung, and skin [3]. Recent evidence has suggested that mature ILCs can be further classified into group 1, 2, and 3 ILCs based on different expression of transcription factors, cell surface markers, and effector cytokines [3]. ILCs are rapid producers of both proinflammatory and regulatory cytokines in response to local injury, inflammation, pathogen infection, or commensal microbiota perturbation [3]. Because most ILCs have been shown to be tissue-resident (with the exception of circulating natural killer (NK) cells) in almost all organs analyzed [5,6], their ability to quickly respond to tissue stress and inflammation underpins their critical role in regulating tissue homeostasis and repair during infection or injury [3]. In this section, we will review the contribution of tissue-resident ILCs to host protection and pathology following viral infection.

### 2.2. Group 1 Innate Lymphoid Cells

Group 1 ILCs consist of both mature NK cells and type 1 innate lymphoid cells (ILC1). They are further defined based on the production of the signature cytokine interferon (IFN)-γ in response to the proinflammatory cytokines interleukin (IL)-12 and IL-18. While both mature NK cells and ILC1 require the group 1 ILC-defining transcription factor *Tbx21* (T-bet) for their development, ILC1 differ from circulating NK cells based on their long-term tissue residency in parabiotic mouse experiments, *Eomes*-independent development, and the expression of the tissue-resident ILC markers CD61 and CD200r1 [3,7]. In experiments studying the kinetics of murine cytomegalovirus (MCMV), influenza, and Sendai virus infection, ILC1 were shown to promote early antiviral responses at the initial sites of infection via rapid production of IFN-γ, well before other known tissue-resident innate and adaptive lymphocytes [7,8]. Furthermore, genetic ablation of ILC1 led to increased MCMV viral load in the liver and increased mortality in the presence of intact NK and T cell responses [7]. These studies collectively suggest a non-redundant and essential host protective role for ILC1 during initial viral immunosurveillance.

In addition to their cell-intrinsic host-protective role early during viral infection, recent studies suggest that ILC1 may also regulate circulating adaptive T cell responses during viral infection. For example, liver ILC1 display enhanced expression of NKG2A compared to liver NK cells, and genetic deficiency of NKG2A leads to increased DC activation and expansion of adenovirus-specific CD8^+^ T cells [9]. During LCMV infection, ILC1-intrinsic increases in PD-L1 expression can suppress LCMV-specific T cell responses following adoptive transfer into sub-lethally irradiated hosts [10]. However, whether ILC1 are specifically required for these responses, and not NK cells, remains poorly defined due to the lack of genetic evidence that endogenous ILC1-specific PD-L1 and NKG2A expression is required for diminished virus-specific T cell responses. Given the antiviral role for ILC1 in other infection models, it will be of interest to further define the distinct viral escape mechanisms that may lead to protective or tolerogenic crosstalk between ILC1 and T cells. It will also be important to define the role of ILC1 responses during secondary heterologous infection since this would represent a more relevant scenario to inform human tissue-resident immune responses and vaccination.

### 2.3. Group 2 Innate Lymphoid Cells (ILC2)

ILC2 are defined by their production of Th2-related cytokines, IL-4, IL-5, and IL-13 in response to parenchymal cell production of IL-25, TSLP, and IL-33 [3]. While the production of type I and type II interferons produced early during viral infection can largely inhibit lung ILC2 proliferation and function in a STAT1-dependent manner [11,12], ILC2 can serve vital tissue repair function following the resolution of influenza infection through the production of amphiregulin (AREG) in response to local IL-33 production [13]. However, whether the described tissue repair role of ILC2 in the lung is redundant with CD4^+^ T cell-derived AREG during influenza will require further investigation [14]. Furthermore, IL-5 derived from lung ILC2 following influenza infection has been implicated in the recruitment and accumulation of eosinophils following viral clearance [15], even though the roles of eosinophils in promoting tissue repair or host pathology in this context are unknown. Although not directly addressed in viral infection models, recent studies have shown that IL-9 produced by ILC2 can prevent sepsis-induced pyroptosis in lung endothelial cells and contributes to the resolution of inflammation during rheumatoid arthritis [16,17]. Thus, while early evidence suggests an integral role for lung ILC2 in tissue repair following viral infection, it will be of interest to further profile the effector cytokines and molecular mechanisms that can induce this process following respiratory infection.

In contrast to the reported host protective roles of lung ILC2, respiratory viral infection can also result in dysregulated type 2 immune responses, which can lead to allergic inflammatory reactions. Infection with influenza virus, rhinovirus, and respiratory syncytial virus (RSV) can all result in ILC2-mediated airway hyperreactivity and inflammation via IL-33 and TSLP production [18,19,20]. Similarly, IL-33 or allergen primed lung ILC2 have been shown to acquire memory-like properties that result in more potent responses to an allergen re-challenge in comparison to primary ILC2 responses [21]. However, whether inflammation-primed ILC2 display enhanced effector function following viral infection is unknown. Since a recent study demonstrated that prostaglandin E2 (PGE_2_) can inhibit IL-33 induced ILC2 cytokine production and subsequent asthmatic readouts in mice [22], it will be of future interest to determine whether certain respiratory viruses regulate the production of PGE_2_ in infected cells to exacerbate ILC2 responses as a viral immunoevasion strategy. Thus, a more complete understanding of the mechanisms of ILC2-related allergic immunopathology during certain types of viral infection will be crucial for the development of new treatment strategies for asthmatic patients.

### 2.4. Group 3 Innate Lymphoid Cells (ILC3)

Group 3 innate lymphoid cells (ILC3) are defined by their expression of the transcription factor retinoid-related orphan receptor γt (RORγt) as well as the production of the cytokines IL-17A and/or IL-22 in response to acute infection. ILC3 play a vital role within the regulation of immunity in mucosal tissues, which play a supportive function during innate immune responses to bacterial infections by contributing to the epithelial barrier integrity and repair [3]. In addition, IL-22 producing ILC3 have been implicated in the control of rotavirus infection within the gut epithelium by stimulating STAT1-dependent interferon stimulated gene (ISG) expression in intestinal epithelial cells [23]. Because of their protective function in barrier tissues, depletion of ILC3 during simian immunodeficiency virus (SIV) infection in rhesus macaques may also contribute to the loss of intestinal mucosal integrity and disease progression [24,25,26]. While these studies collectively support an indirect host-protective role for ILC3 during viral infection, how ILC3 are depleted following SIV infection remains unclear.

While the inflammatory milieu in the gut during SIV infection has been proposed to directly drive RORγt repression and apoptosis in ILC3 [25,26], a recent study has suggested that CD4^+^ T cell deficiency, compounded with dextran sodium sulfate (DSS)-induced gastrointestinal damage, is associated with the loss of ILC3 in the mesenteric lymph nodes of uninfected macaques [27]. Because CD4^+^ T cells are depleted during SIV infection, it will be important to address whether CD4^+^ T cells are required for the survival of mucosal ILC3 during SIV-associated barrier injury since ILC3 homeostasis is normal in *Rag2*^−/−^ mice that lack T cells [28]. Similar to other ILC subsets, chronic activation of ILC3 can also lead to disease progression. ILC3 production of IL-17F during adenovirus infection was shown to lead to optimal effector CD8^+^ T cell responses in the liver, which further exacerbates viral-induced liver hepatitis in mice [29]. Similarly, IL-23 driven ILC3 production of IL-17A and IL-22 during hepatitis B virus (HBV) infection has been shown to be associated with liver fibrosis in human patients, with liver cirrhosis potentially driven by the recruitment of Th17 cells [30,31]. However, the HBV-specific mechanisms that promote IL-23 production in the liver are unknown. Thus, recent studies suggest that ILC3 play both protective and pathologic roles during viral infection, but further research is necessary to determine the molecular mechanisms and contexts that drive ILC3 responses towards protection or pathology during different viral infections.

Taken together, the roles of ILCs during antiviral responses vary based on the phase of the immune response. The roles of ILCs promote protective immunity or pathologic inflammation in certain contexts. ILC1-produced IFN-γ controls early viral replication and may limit the dysregulation of ILC2 and ILC3 homeostatic repair responses. However, exacerbated type I responses might abrogate the beneficial tissue repair functions of ILC2 and ILC3 by directly inhibiting their activation during viral infection. While viral-induced dysregulation of ILC2 responses can lead to exacerbated asthma, ILC3-derived IL-22 can limit airway hyperreactivity [32], which suggests that ILC3 homeostasis may also be negatively impacted during respiratory infections in addition to mucosal infections. Going forward, it will be important to further elucidate how ILC responses are regulated during viral infection in addition to the study of the mechanisms that promote or prevent ILC-induced pathology. A detailed understanding of the interplay between ILCs and circulating and tissue-resident adaptive lymphocytes during viral infection is needed. Because host-virus co-evolution may drive specific escape mechanisms that are unique to each virus, a more complete understanding of how viruses evade or coopt ILC-mediated effector responses should inform novel vaccination strategies.

### 2.5. Unconventional T Cells

Alongside ILCs, tissue-resident “unconventional” T cell subsets play a critical role in the clearance of pathogens with activity that precedes circulating adaptive T cell responses [2]. Unconventional T cell subsets are composed of natural killer T (NKT) cells, mucosal associated invariant T (MAIT) cells, and γδ T cells, all of which express restricted T cell receptor (TCR) sequences [4]. These T cells are termed “unconventional” because of their limited T cell receptor diversity as well as their ability to specifically recognize alternative microbial antigens that are not presented on major histocompatibility complexes (MHC) I and II [4]. Despite limited TCR repertoires, these populations play a critical role within the early antiviral immune response, which mount rapid effector responses as well as mediate downstream adaptive T cell responses. In this section, we will summarize the roles of unconventional T cells during antiviral responses and their potential therapeutic benefits. 

### 2.6. Natural Killer/ Invariant Natural Killer T (NKT/iNKT) Cells

NKT cells are characterized by their specificity to lipid antigens presented by the MHC-I like molecule CD1d, and can be divided into two subsets based upon their TCR alpha chain expression profiles [4]. These consist of type 1 NKT cells, or “invariant” NKT cells, which are duly named because of their invariant TCRα chain (Vα14-Jα18 in mice, Vα24-Jα18 in humans) and restricted range of TCRβ chains. Type 2 NKT cells or “diverse” NKT cells, on the other hand, have a much larger repertoire of TCRα and TCRβ chain pairings [4]. iNKT cells react strongly to α-GalCer and other similarly structured microbial and mammalian antigens by expanding rapidly upon encounter and adapting an effector-like state with potent cytokine production that persists for days after the initial stimulation [4]. While also capable of antigen-specific cytotoxicity [33], these cells act as potent inflammatory cytokine drivers via production of IFN-γ, tumor necrosis factor-α (TNF-α), IL-4, and IL-17. This rapid and diverse cytokine production by iNKT cells can further potentiate innate and adaptive immune responses by activating antigen presenting cells (APCs), which leads to increased protection against infection [2,4,34]. Because of these properties, recent studies have begun studying the use of iNKT agonists during vaccination to enhance B and T cell responses [35].

Although iNKT cell responses are largely protective during the pathogen challenge, the direct and indirect roles of iNKT cells during antiviral responses may be context-specific. During influenza A infection, iNKT cells were found to be sufficient and necessary for host suppression of viral replication and host survival by regulating immunosuppressive myeloid-derived suppressor cells. These results were also corroborated by associative data during human influenza infection, which suggests that iNKT cell activation may antagonize viral infection-associated immunosuppression across species [36]. iNKT cells have also been demonstrated to play a vital role in CD8^+^ T cell activation and expansion, which leads to increased RSV clearance [37]. Furthermore, infection with kaposi sarcoma-associated herpes virus, herpes simplex virus (HSV)-1, vaccinia virus, human papilloma virus (HPV), and human immunodeficiency virus (HIV)-1 all decrease CD1d expression on infected cells, which suggests a shared evolutionary immunoevasion strategy by viruses to circumvent iNKT cell activation [38,39,40,41,42]. While these studies collectively demonstrate important direct and indirect antiviral roles of iNKT cells, they may also play redundant inflammatory roles during other viral infections. Using Jα281-deficient mice lacking NKT cells, a previous study concluded that NKT cells are dispensable for early clearance of MCMV in the spleen and liver. However, enhanced activation of NKT cells with α-GalCer during MCMV reduced viral titers in peripheral organs, which suggests that while the endogenous role of NKT cells may be redundant in controlling MCMV infection, their effector functions can be harnessed to enhance the protective antiviral response [43]. In support of these findings, recent studies have demonstrated that α-GalCer used as an adjuvant during influenza vaccination enhanced cytotoxic CD8^+^ T cell memory responses in mice [44,45]. Furthermore, administration of α-GalCer adjuvant alongside H1N1 swine flu vaccines led to complete inhibition of viral replication within the respiratory tract of piglets [46]. However, despite the reported efficacy of α-GalCer adjuvants in animal models, clinical trials involving patients with chronic hepatitis B and C virus infection have shown no clear clinical benefit to α-GalCer treatment [47,48]. These results raise important questions about the roles of iNKT cells during acute and chronic infections and how the efficacious iNKT agonist treatments would be in these different contexts. Taken together, previous studies collectively support an indirect antiviral role for iNKT cells either through endogenous activation or therapeutic modulation of their activity through α-GalCer stimulation in vivo. Further studies will be necessary to understand the redundant and non-redundant roles of iNKT cells during different types of viral infections due to shared viral immunoevasion mechanisms that suppress iNKT cell activation.

### 2.7. MAIT Cells

Mucosal associated invariant T (MAIT) cells are defined by their reactivity to the MHC-I-like protein MR1, which plays a critical role in both their development and function [49]. MR1 ligands are mainly derivatives from the synthesis pathways of Vitamin B2 (folate) and B9 (riboflavin), which allows MAIT cells to sense a diverse range of microbial metabolites [4]. As such, MAIT cells are generally known for their ability to rapidly and specifically respond to bacterially-infected cells and yeasts [50,51]. Recent analysis of MAIT cell TCR sequences has led to a better understanding of the diversity of microbial metabolites that MAIT cells can detect, but demonstrate a lack of pre-existing specificity for virally-encoded antigens [52,53]. Although unable to directly sense virally encoded molecules, MAIT cells serve important antiviral roles through the potentiation of proinflammatory responses. MAIT cells robustly respond to IL-12 and IL-18 stimulation, which produces IFN-γ, TNF-α, and IL-17 that can enhance innate and adaptive B cell antibody responses [54,55]. In humans, MAIT cells have been implicated in the antiviral response to dengue virus, hepatitis C virus, influenza virus, and Zika virus in response to IL-12 and IL-18 [56,57]. This evidence is further supported by evidence from MAIT cell deficient MR1^−/−^ mice that display increased weight loss and mortality during H1N1 infection [58]. Similar to ILC subsets, MAIT cells are also depleted during HIV-1 infection [59,60], although whether a small number of functional MAIT cells remain is unclear. Irrespective of these inconsistencies, MAIT effector function can be restored upon treatment with IL-7 in vitro [61]. Given that IL-7 signaling drives optimal mucosal ILC2 and ILC3 homeostasis in mice [62], it will be interesting to determine whether HIV-1 infection regulates IL-7 production in mucosal tissues to circumvent tissue-resident lymphocyte responses.

Although MAIT cell responses have been implicated in acute and chronic viral infections, the association between MAIT cell activation and clinical outcomes varies depending on the type of viral infection. For example, MAIT cell activation during HCV infection correlated with positive disease outcomes, but was also associated with disease severity in patients infected with dengue virus [56]. While these studies suggest that MAIT cell activation and potentiation of type I inflammatory responses may be crucial for control of certain viral infections, their responses might also exacerbate virus-induced pathology in certain settings. Because of these observations, it will be of interest to test whether MAIT cell cytokine secretion may contribute to the hyperreactive phenotypes that can be associated with severe influenza infection [63,64]. Use of MAIT cell deficient MR1^−/−^ mouse models in future studies will be able to carefully examine the role of these cells during chronic viral-induced immunopathology.

### 2.8. γδ T Cells

γδ T cells have several distinct subsets that differ between mice and humans regarding antigen reactivity, tissue residency, and TCR usage. They have been implicated in the establishment and regulation of inflammatory responses, tissue homeostasis, and tissue repair [65]. As such, γδ T cells are among the first line of defense within barrier tissues against bacterial, fungal, and viral pathogens. These cells utilize their cytotoxic capacity via granzymes and perforin to destroy infected cells and early production of IL-17 along with IFN-γ and TNF-α to enhance the immune response [66]. γδ T cells typically respond to phosphoantigens derived from intracellular pathogens, but their expanded receptor diversity allows particular subsets to recognize alternative cell-stress induced proteins, which support their function in the identification and elimination of transformed or infected cells [4]. Additionally, some γδ T cells have the ability to recognize CD1d-based lipid antigens, which may be redundant with iNKT cell activity [67] even though the extent remains unclear.

γδ T cells play an important protective role during infection via potent cytotoxic capabilities as well as the ability to regulate recruitment of other immune cells [68,69]. Infection or reactivation of herpesvirus family members in humans is associated with the clonal expansion of activated γδ T cells, which show reactivity against infected cells through the production of IFN-γ and TNF-α to reduce viral replication [70,71]. However, whether expanded γδ T cell clones display specificity to virus-associated molecules or self-induced stress ligands remains unknown. The antiviral role of γδ T cells is further supported by the use of TCRδ^−/−^ mice that display increased mortality during both vaccination and West Nile virus infection [72,73]. Although γδ T cell antiviral activity is consistent across a range of other viruses (influenza, Epstein-Barr, hepatitis B and C) [74,75,76,77], TCRδ^−/−^ mice do not display increased mortality following MCMV infection but adoptively transferred γδ T cells can rescue the survival defect observed in total T cell deficient mice [78,79]. Together, these results suggest that the antiviral role of γδ T cells may be essential or redundant depending on the specific type of viral infection. Additionally, γδ T cell functionality appears to change over the course of primary to chronic HIV infection where initial anti-inflammatory TGF-β-producing cells give way to pro-inflammatory IFN-γ producing γδ T cells [80]. Dysregulation of γδ T cell activation during HIV infection can have serious consequences since pro-inflammatory γδ T cells have been shown to correlate with HIV viral load [81]. Thus, γδ T cell activation can lead to overactive systemic immune activation, which may contribute to disease progression even though further research is required to determine the extent to which this occurs during other chronic viral infections.

γδ T cell deficient mouse models have highlighted the importance of these cells during antiviral responses, which have lead to active research into the use of phospho-antigen treatment to selectively activate γδ T cell responses. Such treatments have proven effective in controlling viral infections with aminobisphosphonate pamidronate-based expansion of γδ T cells leading to reduced disease severity and mortality during H1N1 and H5N1 influenza infection [82]. Additional non-peptidic drugs, such as Phosphostim and Zoledronate, have been implicated in the treatment of HCV, via γδ T cell-mediated reduction of viral replication [74]. Thus, while early studies involving γδ T cell phosphoantigen treatment are promising, the spectrum of γδ T cell functions in acute and chronic viral infections will need to be further tested to fully understand the immunotherapeutic potential of these methods. 

Despite limited T cell receptor diversity, the ability of the unconventional T cell subsets to mount swift and potent antiviral responses through interplay with other innate and adaptive immune cell subsets reinforces their importance during viral infection. Specific knockout mouse models have helped elucidate the redundant and nonredundant roles of unconventional T cells during different viral infections. Furthermore, the ability of unconventional T cells to regulate the adaptive immune response in an adjuvant-like manner will lead to the development of more precise iNKT, MAIT, and γδ T cell-based agonists to improve current vaccination strategies.

## 3. Conclusions

A growing body of evidence suggests that tissue-resident lymphocytes (including ILCs and unconventional T cell subsets) are critical first responders during initial pathogen challenge. During the antiviral response, these cells elicit a broad range of effector mechanisms: from direct suppression of viral replication, potentiation of local and circulating type I inflammatory responses, and tissue repair (Figure 1). While these mechanisms are host protective by design, dysregulation of these responses can lead to exacerbation of the inflammatory response and virus-associated pathology. However, there are still open questions that need to be addressed to more completely understand the role of specific tissue-resident lymphocytes during viral infection. For example, how specific viral infections, such as HIV-1, deplete mucosal-resident lymphocytes remain poorly defined. In addition, the majority of studies have not used cell type specific genetic ablation strategies to profile the endogenous role of tissue-resident lymphocytes during viral infection. Thus, whether redundancy exists between unconventional T cell subsets and ILC responses, and tissue-resident and circulating lymphocyte responses remains unclear for most viral infections (Table 1). Irrespective of these points, a more complete understanding of specific viral immunoevasion strategies may not only elucidate the evolutionary necessity for redundancy in the tissue-resident lymphocyte responses to viral infection, but also inform strategies for the treatment of tissue pathology associated with chronic viral infections. Studies elucidating the mechanisms of tissue-resident lymphocyte responses to viral infection may also prove to be beneficial for therapeutic use in other infectious diseases and cancer.

## Figures and Tables

**Figure 1 viruses-11-00272-f001:**
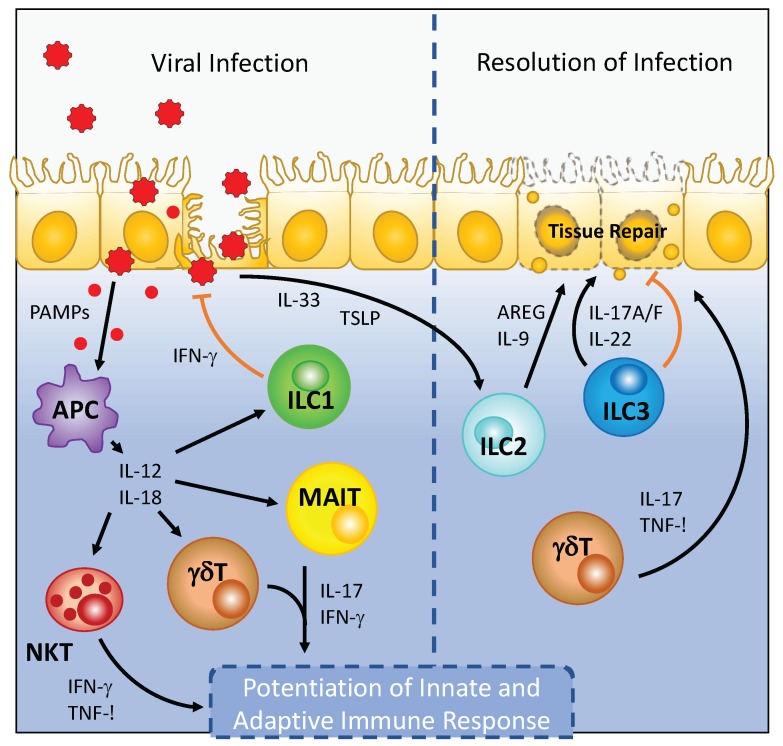
Viral infection leads to the activation of tissue-resident ILCs and unconventional T cell subsets. Recognition of pathogen associated molecular patterns (PAMPs) by antigen presenting cells (APC), which include tissue-resident macrophages and dendritic cells, during viral infection leads to the production of proinflammatory cytokines that include IL-12 and IL-18. IL-12 and IL-18 stimulation of ILC1, NKT, MAIT, and γδ T cells leads to robust cytokine production by these cell types to inhibit viral replication and stimulate subsequent circulating adaptive lymphocyte responses. During the recovery phase of viral infection, the release of IL-33 promotes amphiregulin production by ILC2 to promote tissue repair. Similarly, IL-22 produced by ILC3 can promote barrier integrity and directly limit viral replication to resolve viral infection.

**Table 1 viruses-11-00272-t001:** Protective and pathologic tissue-resident lymphocyte responses to viral infection.

Cell Type	Virus	Protection or Pathogenesis	Genetic Evidence *	Redundancy	Reference
ILC1	MCMV	Protection	Yes	No	[7]
ILC1	Influenza A (H1N1)	Protection	No	?	[8]
ILC1	Adenovirus, LCMV	Protection	No	?	[9,10]
ILC2	Influenza A (H1N1)	Pathogenesis	No	?	[11]
ILC2	Influenza A (H1N1)	Protection	No	?	[13]
ILC2	Influenza A (H1N1)	Protection	No	?	[15]
ILC2	Influenza A (H3N1)	Pathogenesis	No	?	[18]
ILC2	Rhinovirus	Pathogenesis	No	?	[19]
ILC2	RSV	Pathogenesis	No	?	[20]
ILC3	Rotavirus	Protection	No	?	[23]
ILC3	SIV	Protection	No	?	[24,25,26]
ILC3	Adenovirus, LCMV	Pathogenesis	No	?	[29]
ILC3	Chronic HBV	Pathogenesis	No	?	[30,31]
iNKT	Influenza A (PR8)	Protection	Yes	No	[36]
iNKT	RSV	Protection	No	?	[37]
iNKT	MCMV	Protection	Yes	Yes	[43]
iNKT	Influenza A	Protection	No	?	[44,45]
iNKT	Influenza A (H1N1)	Protection	No	?	[46]
iNKT	Chronic HBV, HCV	Neither	No	?	[47,48]
MAIT	Dengue, HCV, Influenza	Both	No	?	[56]
MAIT	Influenza A (H1N1)	Protection	Yes	No	[58]
MAIT	HIV	Protection	No	?	[61]
*γδ*	HHV	Protection	No	?	[70,71]
*γδ*	Vaccinia, West Nile	Protection	Yes	No	[72,73]
*γδ*	MCMV	Protection	Yes	Yes	[78,79]
*γδ*	HIV	Pathogenesis	No	?	[81]
*γδ*	Influenza A (H1N1, H5N1)	Protection	Yes	No	[82]

* Genetic evidence refers to the specific ablation of the indicated cell type in the presence of an intact immune system. ? refers to incomplete evidence for subset redundancy in the referenced study. ILC: Innate lymphoid cell. MCMV: Murine Cytomegalovirus. LCMV: Lymphocytic Choriomeningitis Virus. RSV: Respiratory Syncytial Virus. HBV: Hepatitis B Virus. SIV: Simian Immunodeficiency Virus. iNKT: invariant Natural killer T cell. HCV: Hepatitis C Virus. NKT: Natural killer T cell. MAIT: Mucosal associated invariant T cell. *γδ*: *γδ* T cell. HHV: Human Herpesvirus. HIV: Human Immunodeficiency Virus.

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
