# Peer review of "Tissue-Resident Innate and Innate-Like Lymphocyte Responses to Viral Infection"

_viruses, 2019, doi:10.3390/v11030272_

Round 1

Reviewer 1 Report

This is an outstanding review article that brings together studies on ILC and the control of viral infection. An important contribution to the field. 

Author Response

We thank the reviewer for the positive feedback. 

Reviewer 2 Report

The review by Hildreth et al. compromises current knowledge on the role of ILCs and uncoventional T cells during virus infection. The review is well-written and provides a helpful overview to the topic. The authors represent the opinion that these cells are important during viral infection. The null-hypothesis in this context is not discussed in the review.

The authors summarize specific roles of the cell subpopulations without obvious bias.

A few reports of the role of gd T cells in CMV infection could be included:

10.1038/ni.3686

10.1371/journal.ppat.1004702

10.1371/journal.ppat.1004481

General open questions that could be included in the conlusions/ discussion:

Data on how many ILCs / uncoventional T cells (except NK cells) are present in tissue before and early during infection. What is the effector:target ratio?

A few words on how mouse models used in this research field and how they could be improved.

Author Response

We thank the reviewer for their insightful comments. We have included the proposed references and discussed these studies in the underlined text of the revised manuscript. We have also updated the discussion on the mouse models used and potential improvements for these models in the conclusion section of the revised manuscript.